# Experimental Studies to Test a Predictive Indoor Radon Model

**DOI:** 10.3390/ijerph19106056

**Published:** 2022-05-16

**Authors:** Simona Mancini, Martins Vilnitis, Nataša Todorović, Jovana Nikolov, Michele Guida

**Affiliations:** 1Laboratory “Ambients and Radiations (Amb.Ra.)”, Department of Computer Engineering, Electrical Engineering and Applied Mathematics (DIEM), University of Salerno, 84084 Fisciano, Italy; miguida@unisa.it; 2Institute of Construction Technology, Faculty of Civil Engineering, Riga Technical University, LV1048 Riga, Latvia; martins.vilnitis@rtu.lv; 3Department of Physics, Faculty of Sciences, University of Novi Sad, 21000 Novi Sad, Serbia; natasa.todorovic@df.uns.ac.rs (N.T.); jovana.nikolov@df.uns.ac.rs (J.N.); 4Faculty of Civil Engineering, Riga Technical University, LV1048 Riga, Latvia

**Keywords:** indoor radon, modeling, radon measurements

## Abstract

The accumulation of the radioactive gas radon in closed environments, such as dwellings, is the result of a quite complex set of processes related to the contribution of different sources. As it undergoes different physical mechanisms, all occurring at the same time, models describing the general dynamic turns out to be difficult to apply because of the dependence on many parameters not easy to measure or calculate. In this context, the authors developed, in a previous work, a simplified approach based on the combination of a physics-mathematical model and on-site experimental measurements. Three experimental studies were performed in order to preliminarily test the goodness of the model to simulate indoor radon concentrations in closed environments. In this paper, an application on a new experimental site was realized in order to evaluate the adaptability of the model to different house typologies and environmental contexts. Radon activity measurements were performed using a portable radon detector and results, showing again good performance of the model. Results are discussed and future efforts are outlined for the refining and implementation of the model into software.

## 1. Introduction

Radionuclide ^222^Radon (henceforth, shortly indicated as Radon) is one of the most important sources of exposure to high energy ionizing radiations naturally occurring in the environment. Due to its gaseous state under standard conditions of temperature and pressure and its half-time (around 4 days, roughly), radon can easily be transported by any fluid carrier over a distance from the production source and enter, reach, and accumulate in living places, constituting a potential health risk for occupants.

The chronic long exposure to radon concentrations may turn out to be the second highest cause of lung cancer after smoking [1]. For this reason, over recent decades many efforts have been made to deepen the general understanding of the dynamics, from its entry and accumulation in closed environments to its time and seasonal variations as well. The complexity, due to the large number of parameters and processes responsible for the accumulation of Radon into the environment until it reaches high harmful levels, led to the development and implementation of a large number of different tools for radon risk assessment that include mapping, modeling, etc. [2,3,4,5,6,7,8].

To estimate the individual’s cumulative indoor radon exposure, an assessment of the indoor radon activity concentrations turns out to be a strategic point [9]. For this purpose, physical-mathematical models describing the general accumulation phenomena play a crucial role. They provide not only a tool for estimating the indoor radon concentrations but also information on the understanding and managing of the parameters governing the indoor radon levels according to the site-specific surroundings, the climate, and the features of the building.

In this context, in the framework of a regional project [10] a simplified approach, the S.I.R.E.M. procedure (Simplified Indoor Radon Entry Model), to assess indoor Radon activity concentrations, based on on-site experimental evaluations of Radon emission by building materials and in soil gas, was presented by authors [11].

The main idea of the proposed model is the definition of a room model for each different building type describing the indoor radon accumulation process (the case study here described adds a new typology of ‘reference’ building). In this way, it could be easier to identify buildings typologies more susceptible to high indoor radon concentrations and, also, to easily identify the best mitigation systems according to the sources and main accumulation mechanism. Furthermore, based on this outcome, some software for the assessment of the performance of buildings in terms of indoor radon could be developed and used for real-estate transactions to inform buyers and owners about possible hazards.

Therefore, the present work summarizes new results from experimental studies performed on a case study to test the validity and adaptability of the model and the performance of the building type in terms of indoor radon accumulation.

## 2. Materials and Methods

As more extensively described in [11], S.I.R.E.M. is a tool evaluating indoor radon concentrations based on a set of preliminary experimental measurements from all Radon sources and on the acquisition of other data. To have a real-time picture of the site-specific conditions, the experimental protocol requires real-time measurements by means of an active monitoring system, enabling easy simultaneous acquisition of information about the contribution from the different sources and the cumulative indoor radon concentration.

A continuous radon monitor (CRM), using an ion-implanted solid state detector for the alpha spectrometry of the short-lived radon/thoron progeny, the RAD7 instrument manufactured by DURRIDGE, Inc. (Bedford, MA, USA) [12] was used. The RAD7 CRM has a special internal protocol, the so-called “Radon Sniffer Mode”. In a single data acquisition run, lasting only 30 min, it is able to reconstruct the radon activity concentration taking into accounts only the total counts of alpha particles from the 3-min first radon progeny (6.00 MeV), ^218^Po decay, without any noisy interference from the environmental background gamma radiation or any other ionizing radiation. It also detects, by sniffing, the instantaneous alpha decay of the first thoron daughter (^216^Po) for rapid thoron measurements. The accuracy of the instrument, annually calibrated in a certified laboratory, guarantees a sensitivity of 34 Bq/m^3^ (Minimum Detection Limit, MDL), with a 10% standard deviation (SD), in less than two hours. At the end of each data acquisition run, the detector prints out a complete report.

It is a very versatile instrument largely used in environmental radioactivity investigations [13], since it can perform measurements in water [14], in the soil-gas, and building materials, according to the default protocols and set-ups provided.

### 2.1. Description of the Experimental Site

The experimental site is located in the town of Pompeii. Pompeii is part of the 18 municipalities, called the “Vesuvian towns”, of the metropolitan area of Naples in the Campania Region (south of Italy). Geographically, Pompeii is located in the alluvial plain of Sarno, on the slopes of the ‘Vesuvius Volcanic Park’. From a geological point of view, the pyroclastic subsoil is constituted of sandy pozzolan, medium dense sands, ignimbrite rocks, and limestone rocks. The calcareous substrate is found at depths greater than −30 m.

Radon measurements were performed in a private residential flat, on the second floor of a building located in the urban center (Figure 1). The building, constructed in the 1960s, has six floors above ground. On the ground floor there are business premises, while on the other floors there are residential apartments. The building structure, typical of the architectural Italian tradition of the 1950s and 1960s, is made of a load-bearing masonry, in yellow tuff, along the overall perimeter and reinforced concrete frames and pillars in the central area. The foundation is constituted of pillars on isolated plinths and connecting beams, configuring, from a structural point of view, the so-called ‘mixed structure’. The floor is in cement brick, the internal walls are made of clay bricks.

The test structure is a room in a 100 square meter apartment on the second floor, as stated above. Radon measurements in indoor air, in different conditions, were performed in order to compare experimental results with the calculated ones by the application of the S.I.R.E.M. model. Building materials surface emissions measurements were performed and included in the model too.

Figure 2 shows the structural features (typology) of the test room, of about 38 m^3^, with evidence of the different materials delimiting the room volume and the presence of a window, of about 1.8 m^2^, assuring natural ventilation. There are no heating, cooling, or ventilation systems in the room.

### 2.2. Description of Measurements

The radon measurements were carried out by means of the DURRIDGE RAD7 CRM and in compliance with the internal procedures of the “Ambients and Radiations (Amb.Ra)” Laboratory of the University of Salerno, Italy, ISO 9001:2015 certified. The measurement protocols are described in detail in [11].

Data about environmental conditions (temperature, humidity, presence or not of natural or forced ventilation) and the related photographic documentation were also collected. Measurements of the emission from building materials were performed on the internal wall (made from bricks of clay) and the perimetral wall (in yellow tuff), using the DURRIDGE Surface Emission Chamber (Figure 3a), in order to assess the emissions into the room from all the different building materials [12].

The measurements on building materials refer to radon emissions from surfaces. The ability of the used CRM to count only the polonium-218 decays means that dynamic measurements are clean, and not complicated by long-half-life events radon emission chamber. In a closed-loop configuration, used in this case, the system is first purged, then sealed. Next, the radon concentration within the loop is monitored in SNIFF mode, with short (15 min) cycle times, for a few hours. The initial rate of increase in radon concentration (neglecting the first 15-min cycle), multiplied by the volume of the closed-loop system, gives the rate of radon emission, instead.

In this configuration, the instrument returns, directly, the exhalation from the surface measured in Bq/m^3^, extremely convenient for the model analysis. This measurement configuration provides a more realistic assessment of the indoor radon concentration with respect to the laboratory evaluation of the exhalation rate in samples because, on site, the emission from walls [15] is strongly affected by the presence of plaster, humidity [16], aging [17] ‘walls’ breathability’, etc.

Regarding the indoor radon assessment measurements, in order to detect the maximum reachable level of concentration in the room, measurements were performed after a well-defined period by keeping the door and windows closed. The aim is to estimate radon ingrowth in the room in low ventilation conditions, qualitatively.

In this condition of ‘closed room’, the dominant mechanism of radon entry is the diffusion mechanism, caused by the difference of the radon concentrations in the air and the radon sources (i.e., the building materials). Under equilibrium conditions for the radon accumulation due to the diffusion mechanism of entry with constant ventilation conditions in the room, an equilibrium radon concentration, C_eqD_ is achieved:C_eqD_ = E_D_/λ(1)
where E_D_, expressed as the sum of all the radon fluxes from the various surfaces, is the total rate of diffusion radon entry into the room, λ is the global air exchange rate in the room [18].

Using Equation (1), once the room reached the equilibrium radon concentration, it is possible to estimate the ‘leakage’ of the room, corresponding to the air exchange rate.

Instead, measurements after the phase of opening window aim were performed to simulate occupant’s habits and associate, qualitatively, a value of the ventilation rate factor according to the number and duration of the opening of the windows and door.

In addition, as the RAD7 CRM enables performing radon measurements in two modes, thoron-on and thoron-off mode [12], measurements were made both in the center of the room, where it is reasonable to expect that the thoron activity concentration is practically negligible, and in the thoron-on mode, at a distance of 15 cm from the walls of yellow tuff, since due to the short distance the measured value could be affected by the exhalated thoron, which has a very short half-time <60 s [19] (see Figure 3b).

Indeed, according to its volcanic origin, tuff likely represents, in this case study, the main potential source of radon (see Figure 3d).

No measurements were performed on the floor since the room under investigation is located in a flat on the second floor of a residential multifamily building and there is no possibility of contribution from the soil. Anyway, the area was investigated by authors in previous works and the radon activity concentrations in the soil vary in a class range of 10–20 kBq/m^3^ [10].

### 2.3. Description of the Model

A model was developed by authors for the calculation of the Radon activity concentration in indoor air, taking into account the main parameters affecting the global Radon dynamics, from its generation into the source (soil, water, or building material) to its accumulation in confined spaces. It integrates values of Radon activity concentration measurements on site from soil (*C_US_*; *C_ds_*), building materials (*C_bm_*) and outdoor air (*C_e_*) by means of a Radon detector for real-time and protocol and procedures ad-hoc developed by the authors. Moreover, air exchange rates related to the overall ventilation are considered (*q_i,j_*).
(2)$$C_{i}=\frac{N_{i}}{V_{i}}\lambda_{Rn}=f(C_{US},C_{ds},C_{bm},C_{e},q_{i,j})$$

It is a simplified version of a more complex dynamic theoretical one [4] aimed not to describe in detail the fluid dynamic of the accumulation phenomena, but to reach a level of simplification to describe the phenomena in a reasonably accurate and reliable way for being easily applicable with cheap and easy-to-use tools. The main idea is to include the model in a software describing the performance of a building, in terms of indoor air quality.

One of the major assumptions associated with the proposed simplified model is to treat the building as a single well-mixed zone and to use a steady-state analysis. Similarly, soil response to unsteady driving pressures is not well studied. Both of these assumptions can break down, in real cases, in buildings. Nevertheless, they can be tolerated considering that the study is targeting the one most exposed room in the house and considering that to mitigate Radon entry means evaluating the best solution in the worst case possible (e.g., steady-state conditions with no ventilation).

The model ignores the impact of the local non homogeneities of the soil responsible for Radon migration, based on experimental measurements on site and stack or wind effects in the soil due to the presence of the house. It also does not take into account the description of the dynamics due to the wind and stack effects, important for determining infiltration, because from a radon perspective, the most important of the natural driving forces is the winter stack effect, evaluated by just measuring experimentally the indoor and outdoor temperature and pressure.

All this assumed indoor accumulation depends on the balance between the entry processes (radioactive decay) related to different sources (*C_US_*; *C_ds_*; *C_bm_*, *C_e_*) and removal ones (air exchanges, *q_i,j_*).

Focusing the study, considering only the most exposed room of the house as a well-mixed, single-zone box, the model can be applied considering the contribution of the sources of Radon, experimentally measured, and the indoor radon concentration in order to understand how it accumulates according to different levels of air leakage from indoor/outdoor air (*q_ij_*). Therefore, air exchange is not physically measured but the intent is to use the model to associate different level of natural ventilation, in terms of number of air exchange per hour.

## 3. Results and Discussion

The results of the measurements campaign performed are reported in Table 1, Table 2, Table 3 and Table 4.

All these measurements were performed not only for monitoring purposes but for proposing the use of a model describing the radon dynamics for the calculation of the global air exchange in the room. Since the model was already tested in a previous study [11] in three test rooms located on the basement floor of residential buildings representing three typical typology of buildings in rural areas of the Campania region (an old detached house, completely built in tuff with direct foundations; a semidetached house, originally in tuff, renewed with brick and concrete; a new detached house in concrete and bricks), in this paper, a typical typology of building in an urban area has been considered instead. 

Since, in this case, soil does not contribute to the indoor radon accumulation, the algorithm is made up of data reported in Table 5, describing the diffusion entry into a room from a radon source, basically.

The indoor radon concentration in various conditions of ventilation were measured with a CRM, as above described, and V_i_, S_bm_, and C_bm_ were measured, too. E_D_ can be calculated as the sum of radon fluxes through the surfaces and λ could be indirectly calculated through the model by inputting the indoor radon concentration, C_i_, experimental values.

The results obtained by the analytical model are reported in Table 6.

Since the indoor radon concentration at the equilibrium is 0.31 ± 0.11 kBq/m^3^ the corresponding calculated value of λ, equal to 0.23, could be considered to be the ‘leakage’ of the room corresponding to the air exchange rate through walls, doors, and windows. 

## 4. Conclusions

In this paper, an experimental study to test an indoor radon model to indirectly calculate the air exchange in a room was presented. Radon activity measurements were performed using a portable radon detector. The measurements included indoor and building materials. Results show how not negligible radon indoor concentration could be found in confined spaces due only to the presence of emitting building materials. Generally, apartments located on the upper floors are considered to be not very exposed to Radon, since the soil is the main source. For this reason, the national legislation recommends monitoring residential and public places located on the basement and ground floors. However, some building materials could represent a significative source of Radon affecting the indoor environment [20]. Therefore, the monitoring should be not only limited to basement and ground floors but also extended to other floors in some typologies of buildings.

The data were used also to test a radon model developed by authors in previous works for the indirect measurement of the ventilation rate. The model seems to return reasonable values, typical of natural ventilated indoor environments. Future efforts will be made in order to experimentally measure the air exchange rate to compare the calculated values with the measured one. Indeed, these experimental studies are part of a wider work aimed to investigate more in depth the radon accumulation dynamic and to develop a model able to simulate the performance of a building from an indoor air quality perspective and, finally, use predictive radon model to indirectly obtain values of the ventilation rate in a room.

## Figures and Tables

**Figure 1 ijerph-19-06056-f001:**
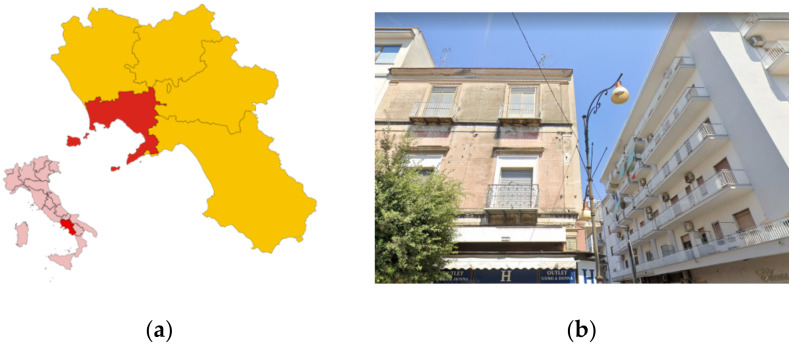
Experimental site location: (**a**) geographical and (**b**) urban context.

**Figure 2 ijerph-19-06056-f002:**
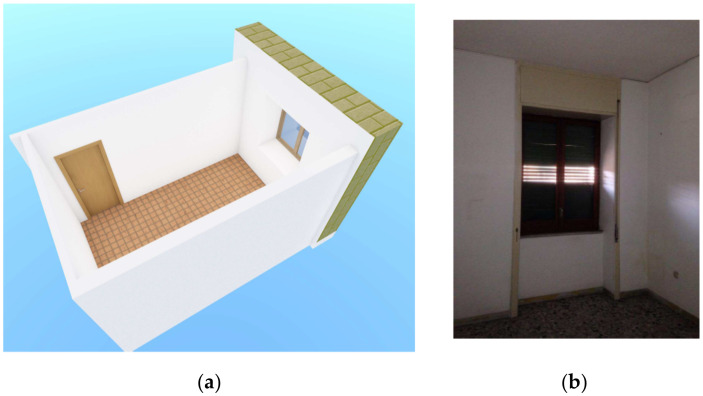
(**a**) 3D drawing and photo (**b**) of the test room.

**Figure 3 ijerph-19-06056-f003:**
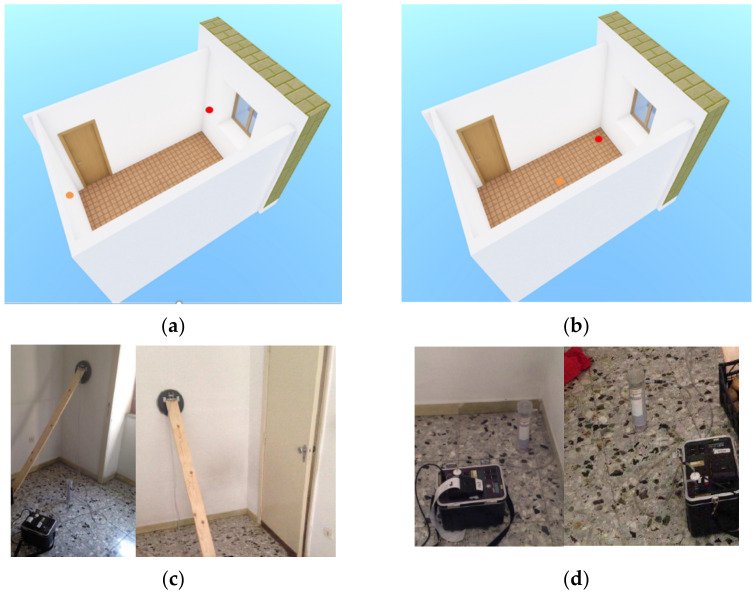
3D drawing with evidence on the localization of the instrument (red dot) for measurements on (**a**) walls and (**b**) indoor air. Set up of the instrument: wall (**c**) and indoor air (**d**) measurements.

**Table 1 ijerph-19-06056-t001:** Soil gas measurements data.

Protocol Name	Duration [hh:mm]	^222^Rn Activity Concentration [kBq/m^3^]	Outdoor Weather Condition	
			Temperature	Humidity
			[°C]	[%]
GRAB	00:30	15.20 ± 90	21	69
GRAB	00:30	9.22 ± 51	25	68

**Table 2 ijerph-19-06056-t002:** Building materials measurements data.

Protocol Name	Duration [hh:mm]	^222^Rn Activity Concentration [kBq/m^3^]	Description of the Wall
BM	02:00	0.2 ± 0.06	50 cm yellow tuff covered by 2 cm of plaster
BM	02:00	0.06 ± 0.03	10 cm clay brick covered by 1.5 cm of plaster

**Table 3 ijerph-19-06056-t003:** Indoor measurements data.

Protocol Name	Duration [hh:mm]	^222^Rn Activity Concentration [kBq/m^3^]	^220^Rn Activity Concentration [kBq/m^3^]	Outdoor Weather Condition	
				Temperature	Humidity
				[°C]	[%]
GRAB (thoron on)	00:30	0.09 ± 0.04 ^1^	0.08	4	31
GRAB(thoron on)	00:30	0.05 ± 0.02 ^2^	0.05	32	55

^1^ reached by taking closed door and windows in the room for 48 h. ^2^ reached 30 min after opening the window.

**Table 4 ijerph-19-06056-t004:** Indoor measurements data.

Protocol Name	Duration [hh:mm]	^222^Rn Activity Concentration [kBq/m^3^]
GRAB	00:30	0.30 ± 0.10 ^1^
GRAB	00:30	0.18 ± 0.05 ^2^

^1^ reached by taking closed door and windows in the room for 1 month; ^2^ reached 30 min after opening the window.

**Table 5 ijerph-19-06056-t005:** Input parameters of the model.

Symbol	Description
V_i_	Volume of the room
S_bm_	BM surface area
C_i_ C_eq_ E_D_	indoor radon concentration equilibrium indoor radon concentration total rate of diffusion radon entry
C_bm_	BM radon concentration
λ_rn_	Radon decay constant
λ	ventilation rate factor

**Table 6 ijerph-19-06056-t006:** Calculation of the ventilation rate factor, λ by converging indoor ^222^Radon activity concentration by experimental measurements and model.

^222^Rn Activity Concentration ^1^ [kBq/m^3^]	^222^Rn Activity Concentration ^2^ [kBq/m^3^]	λ [h^−1^]
0.31 ± 0.11	0.30 ± 0.09	0.23
0.18 ± 0.48	0.20 ± 0.05	0.39
0.09 ± 0.045	0.09 ± 0.03	0.74
0.05 ± 0.02	0.06 ± 0.02	1.20

^1^ from experimental measurement; ^2^ from model.

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
