# Peer review of "Experimental Studies to Test a Predictive Indoor Radon Model"

_ijerph, 2022, doi:10.3390/ijerph19106056_

Round 1

Reviewer 1 Report

As specified L56-60, this manuscript aims to summarize results for testing of the validity of an indoor radon model. However, as I understand the text, results are only presented for one room in one house (so there seems not to be not much to summarize) and the measurements are even simple spot measurements (30 min duration) of radon-222. The key results are in Table 6 where measured and modelled indoor radon levels are compared for four different air-exchange rates (0.1, 0.3, 0.4, and 0.6 h-1). It is, however, unclear how air-exchanges were measured. In fact, the table says that the ventilation rate factor was assigned, so perhaps no measurements were performed? It is also unclear in what sense the radon concentration was measured in the building material. Normally, it is of no interest to know the concentration in Bq/m3 in the building material (the exhalation rate in Bq pr. m2 pr. s can, however, be relevant). It is stated L161 that no radon entered from the soil, but it is unclear how this conclusion was reached, and it is not clear how the study can then represent any sort of interesting (critical) test case for an indoor radon model!

Many details (section 2.1) about the site are provided although the relevance of this information is unclear.

The contribution from building materials is normally relatively constant and independent of the air-exchange rate, and if the outdoor radon concentration is low,  the radon concentration should be inversely proportional to the air-exchange rate (cf. Fig. 4). This can hardly be any surprise. This has been demonstrated in many studies, see for example Andersen et al.: Radon and Natural Ventilation in Newer Danish Single-Family Houses Indoor Air 1997; 7: 278-286 for an example of combined measurements of indoor radon and air-exchange rate in 117 buildings.

Table 6 is similar to what was shown as Table 2 and 3 in ref. [11] with the same first author.

In conclusion, this work includes very limited data (short-term measurements in one room in one house), with no solid information or measurement  of air exchange or radon entry rate. The work does not include an interesting scientific hypothesis or any critical evaluation of anything.

Author Response

Please, see the attached file

Reviewer 2 Report

The manuscript number ID: ijerph-1601229 is referred to an application of a simplified approach, developed in a previous work and based on the combination of a physics-mathematical model and on-site experimental measurements, on a new experimental site, with the aim to evaluate the adaptability of the model to different house typology and environmental context. Radon activity measurements have been performed by using a portable radon detector and results are discussed as future efforts outlined for the refining and implementation of the model into a software.

The paper is well-organized. It contains original results and the overall presentation is convincing. The number of references must be increased.

Author Response

Please, see the attached file

Reviewer 3 Report

This manuscript presents an interesting study on radon with experimental values and a predictive model for indoor radon.

There are some comments detailed below:

- At some point, preferably at the beginning of the manuscript, it should be mentioned that "radon" in the text refers to the radionuclide 222Rn. 

- Please, check the spelling of the manuscript in general. Moderate English changes required.

- Lines 31, 35, 43, 45, 76, 86, 107 and 129: check spelling.

- Line 52 and 78: S.I.R.E.M. or SIREM? And please, tell the meaning if it’s of interest.

- Materials and Methods: Please add some information about CRM (DURRIDGE) calibration/verification and also any other quality assurance about radon measurements.

- Line 117: check parentheses

- Table 3: better explain the conditions of the two situations.

- Table 4: there is a footnote (1) without reference in the content.

- Table 6: units for ventilation rate (h-1?). Please add uncertainties for Radon in both experimental and model values.

- Renumber “Conclusions” section to 4.

- It is not clear what is measured in building materials: radon exhalation?

- It would be good to detail what this work contributes with respect to the previous one: Reference [11]. Any improvement in the mathematical model?

Author Response

Please, see the attached file

Reviewer 4 Report

The manuscript "Experimental studies to test a predictive indoor radon model" presents a study comparing physical measurements of Rn to those simulated by model in the residential setting.

Unfortunately, this work does not meet the requirements of a scientific work in terms of novelty. The authors did not present any new idea or methodology and carried out the measurement in only one place.

Therefore, I do not recommend this article for publication. 

Author Response

Please, see the attached file

Round 2

Reviewer 4 Report

The article has been significantly improved, but the comparison of measured and modeled data should be discussed in more detail. The authors adopted the ventilation coefficient based on the energy efficiency standard without unambiguous measurements and evidence. For this reason, the comparison of data is subject to high uncertainty due to the rate of ventilation.

I propose to use the RAD7 data to determine ventialtion rate based on radon emission rate (lines 143-150), descibed (see eq. 9) in the paper Yarmoshenko et al., "Model of radon entry and accumulation in multi-flat energy-efficient buildings", DOI: 10.1016/j.jece.2021.105444

Author Response

Please, find the attached file.
